# A General Constraint for Gaussian Latent Variables

## Abstract

Encoder-based generative models fundamentally rely on the structure of their latent space to achieve high-quality image reconstruction, generation, and semantic manipulation. In latent spaces, a multivariate Gaussian distribution is often desirable due to its closure under linear transformations. To approximate this, most existing methods impose a standard Gaussian prior via Kullback-Leibler (KL) divergence, which assumes independence among latent components. However, real-world latent representations typically exhibit strong internal correlations, rendering the independence assumption inadequate. In this work, we apply random projection theory to analyze how latent representations differ from a target multivariate Gaussian distribution. We prove that the normalized third absolute moment in low-dimensional subspaces effectively quantifies such deviations. Building on this result, we propose a regularization method that encourages the latent space to align with a multivariate Gaussian distribution without independence assumption across dimensions. The method is compatible with a wide range of encoder-based architectures and introduces no additional computational overhead. We validate the effectiveness of our method through extensive experiments across diverse models. The results consistently show improvements in generation quality, semantic editability, and alignment with the target latent distribution, demonstrating the practical value of the proposed regularization.

## 1 Introduction

Encoder-based models have become central to image generation, editing, and reconstruction. By mapping high-resolution images to low-dimensional latent representations, they enable efficient sampling and controllable semantic editing. This latent encoding approach serves as the foundation for models such as variational autoencoders (VAEs) (Kingma et al., 2013), vector-quantized autoencoders (VQ-VAEs) (Van Den Oord et al., 2017), and encoder-based GAN inversion methods (De Souza et al., 2023). These models have demonstrated strong performance across a broad range of tasks in visual synthesis and analysis.

The probabilistic structure of the latent space is crucial to encoder-based generative models. A Gaussian distribution is particularly useful due to its closure under linear transformations, which ensures that operations such as interpolation and attribute editing remain within the same distribution. However, the latent distribution approximates a Gaussian only in sufficiently low-dimensional settings. In practice, enforcing such low-dimensional representations requires a high compression ratio, which inevitably leads to information loss and degrades both reconstruction accuracy and generation quality. This leads to a fundamental trade-off: lower-dimensional representations better align with the Gaussian prior but limit representational capacity, higher-dimensional representations preserve more information but deviate from the Gaussian distribution, which is required for efficient sampling and manipulation in the latent space. To address this, many encoder-based models attempt to impose Gaussian constraints in the latent space without aggressive compression.

A standard solution is to impose a Gaussian prior on the latent space by minimizing Kullback-Leibler (KL) divergence, as done in variational autoencoders (VAEs) and latent diffusion models (LDMs). This regularization implicitly assumes statistical independence among latent components. However, this assumption rarely holds in practice. Latent representations learned from real-world data often exhibit strong internal correlations, reflecting underlying semantic or spatial structure (Ballé et al., 2018). Such independence as-

sumptions often fail in practice, especially in high-resolution image domains where semantic entanglement is prevalent. This mismatch not only limits the expressiveness of learned representations, but also severely degrades generative quality and latent controllability.

A straightforward approach to solving this mismatch is to estimate the latent distribution and compute the KL divergence with respect to a target Gaussian prior. However, accurate density estimation in high-dimensional spaces is challenging, and computing the KL divergence under such conditions is prohibitively expensive. To overcome this limitation, we propose a Gaussian linear constraint that does not rely on the explicit form of the latent distribution. Specifically, we use the random projection theorem to evaluate how closely a projected latent vector approximates a Gaussian distribution. This allows us to evaluate distributional properties without the need for explicit density modeling. Building on these theoretical results, we propose a Gaussian regularization term that can be seamlessly integrated into the training objectives of various encoder-based models.

We evaluate our Gaussian regularization method on a wide range of generative models, including VAEs, VQ-VAE-2 (Razavi et al., 2019), latent diffusion models (Rombach et al., 2022), and GAN inversion frameworks (Tov et al., 2021). Experiments on diverse datasets, including facial images (Liu et al., 2015; Karras et al., 2019), objects (Yu et al., 2015), and facial expressions (Langner et al., 2010), consistently demonstrate improvements in sample quality, alignment with the target Gaussian distribution, and semantic editability, all without increasing training cost. These results highlight the generality, efficiency, and practical effectiveness of our regularization approach.

We summarize our contributions as follows:

- **Theoretical foundation:** We establish a rigorous framework for analyzing latent distributions using sparse random projections. Building on the multivariate central limit theorem and Berry–Esseen-type bounds, we show that the deviation from a multivariate Gaussian distribution can be quantified by a normalized third absolute moment.

- **Lightweight Regularization Design:** We propose a differentiable regularization term with negligible computational cost. It requires no additional parameters, architectural changes, or independence assumptions. The method can be seamlessly applied to a wide range of encoder-based models to encourage Gaussian structure in the latent space without sacrificing image fidelity.

- **Experimental validation:** We evaluate the proposed method on VAEs, VQ-VAEs, latent diffusion models, and GAN inversion frameworks. Experimental results show consistent improvements in generation quality, better alignment of latent distributions with Gaussian priors, and enhanced image editability.

## 2 Theoretical Foundations

### 2.1 Random Projections for Latent Distribution Analysis

In practice, estimating high-dimensional densities is unreliable due to the curse of dimensionality and the complexity of joint distributions. Therefore, assessing whether high-dimensional latent vectors approximate a multivariate Gaussian distribution is statistically challenging. A classical result in probability theory states that a random vector is Gaussian if and only if all of its linear projections are Gaussian. This implies that if the latent space approximates a Gaussian distribution, then its projections in arbitrary directions should also exhibit Gaussian properties. Conversely, deviations in projected vectors indicate latent structure or asymmetry that violates Gaussian assumptions. Building on this theory, we apply random linear projections to analyse the distribution of latent vectors across multiple projection directions.

To evaluate the latent distribution without introducing external Gaussian characteristics, we avoid using traditional dense random projections with Gaussian entries, which tend to bias the projected data toward Gaussian. Instead, we adopt very sparse random projection matrices with non-Gaussian entries, as proposed by Achlioptas (2003) and extended by Li et al. (2006). Each entry $r_{ij}$ is sampled independently from a symmetric ternary distribution with a sparsity parameter $s$ that controls the proportion of nonzero values:

$$r_{ij} = \sqrt{s} \begin{cases} +1 & \text{with probability } \dfrac{1}{2s} \\ \;\;\,0 & \text{with probability } 1 - \dfrac{1}{s} \\ -1 & \text{with probability } \dfrac{1}{2s} \end{cases}, \tag{1}$$

where $s \in \{1, 3\}$ or any larger positive integer. This construction preserves key statistical properties necessary for theoretical analysis: $\mathbb{E}r_{ji} = 0$, $\mathbb{E}r_{ji}^2 = 1$, $\mathbb{E}\,|r_{ji}|^3 = \sqrt{s}$, $\mathbb{E}r_{ji}^4 = s$, and $\mathbb{E}\,[r_{ji}r_{j'i'}] = 0$, $i \neq j$ or $i' \neq j$. These sparse projections preserve pairwise distances in expectation while avoiding the artificial Gaussian bias (Li et al., 2006). Geometrically, this projection embeds latent vectors into multiple orthogonal random directions, consistent with the Gaussian test theory for high-dimensional random vectors. This insight forms the theoretical foundation of our regularization design.

## 2.2 Projection Setup

Let $x_0 \sim p_{\text{data}}(x_0)$ denote a real image sampled from the data distribution, and let $E_\phi$ be the encoder of a given model. The encoder maps $x_0$ into a latent vector $\boldsymbol{z} = E_\phi(x_0) \in \mathbb{R}^m$. For a batch of $n$ samples, this yields a latent matrix $\mathbf{Z} = \left(\boldsymbol{z}_1^T, \boldsymbol{z}_2^T, \ldots, \boldsymbol{z}_n^T\right)^T \in \mathbb{R}^{n \times m}$. We assume the latent distribution has finite fourth-order moments.

To characterize its latent distribution, we adopt a general Gaussian prior $\mathcal{N}(\mathbf{0}, \boldsymbol{\Sigma})$, where $\boldsymbol{\Sigma} = \boldsymbol{\Sigma}_{q(\boldsymbol{z})} \in \mathbb{R}^{m \times m}$ is the empirical covariance matrix of the latent samples. We then assess how closely the latent distribution aligns with this Gaussian by analyzing its properties under sparse random projections. Specifically, we construct a projection matrix $\mathbf{R} \in \mathbb{R}^{m \times l}$, whose entries follow the ternary distribution defined in Section 2. The latent matrix is then projected as:

$$\boldsymbol{Z}' = \frac{1}{\sqrt{l}} \boldsymbol{Z} \boldsymbol{R}, \qquad \boldsymbol{Z} \in \mathbb{R}^{n \times l} \tag{2}$$

Rather than explicitly estimating the density of the full latent distribution, we measure its deviation from a multivariate Gaussian using a normalized third-order moment statistic (introduced in Section 3), which provides a tractable and differentiable objective.

## 2.3 Approximation Bound

We now analyze small-sample properties using sparse random projections. To quantify deviations from a Gaussian distribution at finite dimensionality $m$, we derive a Berry–Esseen-type bound that characterizes the approximation error.

**Theorem 1.** *(Berry-Esseen type bound)*

*Let $\boldsymbol{w}_i \sim \mathcal{N}(\mathbf{0}, \boldsymbol{\Sigma}')$ be a Gaussian vector with the same covariance as $\boldsymbol{z}_i'$. Then for any convex Borel set $B \in \mathbb{R}^l$, the deviation between the distributions of $\boldsymbol{z}_i'$ and $\boldsymbol{w}_i$ is bounded by:*

$$|P\{\boldsymbol{z}_i' \in B\} - P\{\boldsymbol{w}_i \in B\}| \leq C(m)e(l) \cdot \frac{\sum_{j=1}^m |z_{i,j}|^3}{\left(\sum_{j=1}^m z_{i,j}^2\right)^{\frac{3}{2}}} \tag{3}$$

*where $C(m)$ is a constant only relying on $m$, and $e(l) = \mathbb{E}\left[\sum_{k=1}^l r_{jk}^2\right]^{3/2}$ is equivalent for any $\{j = 1, \cdots, m\}$.*

Recent work (Raič, 2019) shows that $C(m)$ can be upper bounded by $42m^{1/4} + 16$ under certain assumptions. The right-hand side of the inequality defines a normalized third absolute moment:

$$\beta(\boldsymbol{z}_i) = \frac{\sum_{j=1}^m |z_{i,j}|^3}{\left(\sum_{j=1}^m z_{i,j}^2\right)^{\frac{3}{2}}} \tag{4}$$

This bound quantifies how closely a projected latent vector approximates a Gaussian distribution. Notably, $\beta\left(\boldsymbol{z}_i\right)$ differs from conventional skewness, as it captures the third absolute moment rather than the signed third moment. It provides a tractable, interpretable, and differentiable statistic for evaluating how far a latent vector deviates from Gaussian structure after random projection. This statistic serves as the core metric in our proposed regularization framework.

# 3 A General Gaussian Constraint

## 3.1 Moment-Based Regularization Design

Building on the theoretical results in Section 2, we propose an interpretable and differentiable regularization term that quantifies and penalizes deviations from a Gaussian distribution in the latent space.

Given a batch of latent vectors $\{\boldsymbol{z}_1, \boldsymbol{z}_2, \ldots, \boldsymbol{z}_n\}$ with each $\boldsymbol{z}_i \in \mathbb{R}^m$, we define the regularization loss as:

$$\mathcal{L}_{G-reg} = \frac{1}{n}\sum_{i=1}^{n} \frac{\frac{1}{m}\sum_{j=1}^{m}|z_{i,j}|^3}{\left(\frac{1}{m}\sum_{j=1}^{m}z_{i,j}^2\right)^{\frac{3}{2}} + \epsilon} \tag{5}$$

where $\epsilon$ is a small positive constant for numerical stability. This term penalizes the average normalized third absolute moment $\beta(\boldsymbol{z}_i)$. As shown in Section 2, it reflects the deviation from a Gaussian distribution under random projections. A lower value of $\mathcal{L}_{G-reg}$ indicates that the latent vectors are closer to a multivariate Gaussian distribution in projected subspaces.

In variational autoencoders and their various extensions, the latent prior is typically assumed to be a standard Gaussian. In contrast, we consider a general Gaussian prior with arbitrary covariance, reflecting the empirical statistics of the learned latent space. Rather than enforcing a standard prior via KL divergence, we substitute it with our proposed regularization term.

$$\mathcal{L}_{total} = \lambda_{l_2}\mathcal{L}_2 + \lambda_{prec}\mathcal{L}_{perc} + \cancel{\mathcal{L}_{KL}} + \lambda_{G-reg}\mathcal{L}_{G-reg} \tag{6}$$

where $\mathcal{L}_2$ is the pixel-wise reconstruction loss, and $\mathcal{L}_{\text{perc}}$ is a perceptual similarity loss.

The proposed regularization is also applicable to non-probabilistic encoder-based models such as GAN inversion frameworks (Tov et al., 2021) and vector-quantized autoencoders (Van Den Oord et al., 2017). In these cases, it is directly added to the original training objective:

$$\mathcal{L}_{total} = \lambda_{origin}\mathcal{L}_{origin} + \lambda_{G-reg}\mathcal{L}_{G-reg} \tag{7}$$

where $\mathcal{L}_{\text{origin}}$ denotes the model's original training loss, such as the reconstruction or adversarial loss. Our regularization method serves as a general regularizer, introducing statistical constraints without modifying the model architecture. In summary, the proposed regularization promotes Gaussian structure in the latent space and can be seamlessly integrated into a wide range of encoder-based generative models without architectural modification.

## 3.2 Computational Complexity Analysis

The proposed regularization term involves computing the normalized third absolute moment for each latent vector. This requires two summations and one power operation per sample, resulting in a total computational complexity of $\mathcal{O}(nm)$ for a batch of size $n$ and latent dimension $m$. All operations are element-wise and parallelizable.

In contrast, the standard KL divergence used in variational autoencoders has the following form:

$$\mathcal{L}_{\text{KL}} = \frac{1}{2n}\sum_{i=1}^{n}\sum_{j=1}^{m}\left(\mu_{i,j}^2 + \sigma_{i,j}^2 - \log(\sigma_{i,j}^2) - 1\right) \tag{8}$$

which also has complexity $\mathcal{O}(nm)$. However, computing this loss requires the encoder to output both a mean vector $\boldsymbol{\mu}$ and a variance vector $\sigma^2$, typically via two parallel linear layers. In contrast, our method is applied directly to the latent matrix, without requiring variance estimation, thereby reducing architectural complexity. In summary, the proposed regularization captures statistical dependencies among latent dimensions while maintaining the same $\mathcal{O}(nm)$ complexity as standard KL divergence, offering a tractable and architecture-agnostic alternative.

# 4 Experiments

To evaluate the effectiveness of the proposed Gaussian regularization, we conduct experiments across diverse encoder-based generative models. In each setting, we compare model performance with and without the regularization term, focusing on its effect on latent distribution alignment and generation quality. Section 4.1 confirms the effectiveness of our method by projection experiments on synthetic Gamma distributions. Section 4.2 compares the statistical properties in latent spaces for a standard autoencoder with and without the proposed Gaussian regularization. Section 4.3.1 investigates the impact of the regularization on reconstruction fidelity and random sample quality in autoencoders and variational autoencoders. We further extend the analysis to discrete latent spaces using vector-quantized autoencoders. Section 4.3.2 evaluates the method within latent diffusion models (LDMs), a state-of-the-art framework for high-resolution image generation. Section 4.3.3 applies the regularization to a StyleGAN2 inversion model, demonstrating improved editability and consistency in the latent space. Section 4.4 assesses the perceptual quality of the outputs by user studies. Section 4.5 evaluates the robustness of our method under different regularization strength, batch size and generation initialization.

## 4.1 Controlled Validation on Synthetic Gamma Distributions

To obtain a clearer and more controlled understanding of the behavior of the proposed Gaussian regularization (GReg), we design an experiment using synthetic data. This controlled setting enables us to isolate the statistical effect of GReg without the confounding factors inherent in learned representations.

We generate a series of data matrices by sampling from Gamma distributions with varying shape parameters. Specifically, the rate parameter is fixed at $\lambda = 1$, while the shape parameter $\alpha$ varies over the set $\{0.02, 0.05, 0.08, 0.1, 0.2, 0.5, 1, 5, 10\}$. For each $\alpha$, we construct a 100-dimensional dataset in which every dimension is independently and identically distributed according to $\text{Gamma}(\alpha, \lambda)$. Each dataset contains 10,000 samples, ensuring stable and representative empirical statistics.

For each shape parameter, we compute $\mathcal{L}_{G\text{-reg}}$ and apply a random projection from 100 dimensions to a single dimension, following the procedure described in Section 2. The projected samples are then used to construct histograms, which are compared against the probability density function of a corresponding normal distribution. As illustrated in Figure 1, we observe that as the $\mathcal{L}_{G-reg}$ decreases, the histogram of the projected samples more closely aligns with the normal reference curve. This finding quantitatively confirms that $\mathcal{L}_{G\text{-reg}}$ serves as an effective Gaussian constraint.

## 4.2 Latent Distribution Diagnostics with Gaussian Regularization

This section compares the statistical properties in latent spaces for a standard autoencoder with and without the proposed Gaussian regularization. To obtain a comparable test-field, we fix the computational resources to a single RTX 4090D for all experiments in this section. We apply regularization constraints in the latent space of autoencoder. Both models follow the architecture introduced in Hou et al. (2017) and are trained for 40 epochs on the CelebA-64 dataset. All training settings, including learning rate, batch size, and optimizer, are held identical to ensure a fair comparison. The only difference lies in the inclusion of our regularization term.

We randomly sample 1,000 validation images and extract their 100-dimensional latent codes via the trained encoders. To visualize the latent space structure, we project these vectors into two dimensions using Principal Component Analysis (PCA) and t-Distributed Stochastic Neighbor Embedding (t-SNE) (Van der Maaten &

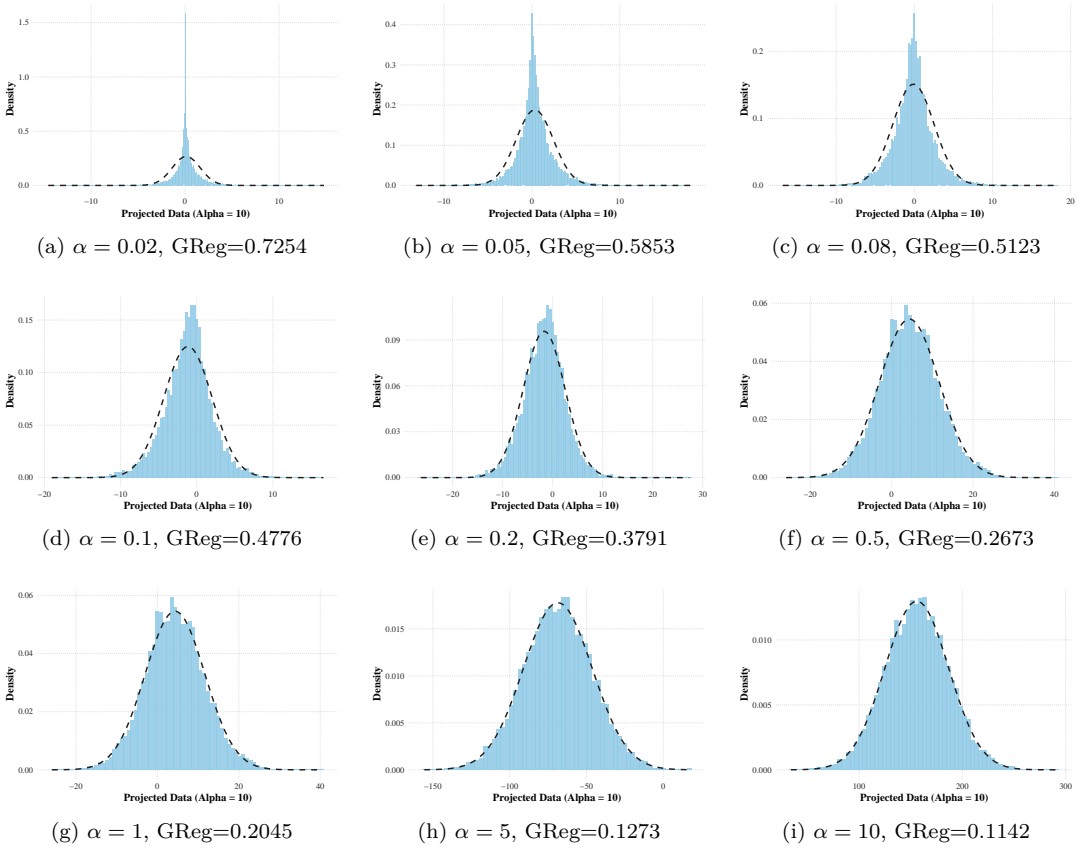

Figure 1: **Controlled validation of $\mathcal{L}_{G-reg}$ on synthetic Gamma distributions.** Each subfigure corresponds to a dataset generated from a Gamma$(\alpha, \lambda = 1)$ distribution with a different shape parameter $\alpha$. Histograms (blue) show the distribution of samples after random projection from 100 dimensions to 1 dimension, following the procedure described in Section 2. The black dashed line represents the probability density function of the corresponding normal distribution. As $\alpha$ increases, the $\mathcal{L}_{G-reg}$ decreases, and the projected distribution aligns more closely with the Gaussian reference, illustrating the theoretical link between $\mathcal{L}_{G-reg}$ values and multivariate Gaussian distribution.

Hinton, 2008). The projection results are shown in Figure 6. PCA is a linear transformation method that extracts the principal components of the data. Due to the closure property of Gaussian distributions under linear transformations, the PCA projection of a high-dimensional Gaussian distribution remains Gaussian in lower dimensions. In particular, a multivariate Gaussian projected to two dimensions should exhibit an elliptical shape. As shown, the 2D PCA projection of the latent vectors from the standard Autoencoder does not display a clear Gaussian-like structure. In contrast, the regularized AE yields a compact elliptical distribution, indicating that the regularization encourages the latent space to better approximate a Gaussian distribution. In addition, t-SNE is a nonlinear dimensionality reduction algorithm designed for visualization by capturing local neighborhood structures. A more compact t-SNE result implies greater similarity in the feature distribution among samples. As illustrated, the regularized AE achieves a more compact 2D layout compared to the standard AE, suggesting that samples in the latent space share similar distributions.

To quantitatively assess the statistical properties of the latent distributions, we compute the skewness and excess kurtosis of the latent vectors from each model, which respectively measure the asymmetry and tail heaviness of the distributions. In addition, we also analyze the PCA-projected representations. Furthermore, we perform the Mardia test, a multivariate normality test, on both the original latent vectors and their PCA projections to evaluate how well the latent representations match a multivariate Gaussian distribution. The

results are shown in Table 1. Compared to the standard AE, the regularized AE shows lower excess kurtosis in both the original and PCA-projected latent spaces. Although the regularized AE has slightly higher skewness in the low dimensional space of PCA projection, its skewness is closer to a Gaussian distribution in the high-dimensional space. It also achieves a lower Mardia test statistic, indicating that the regularized latent space more closely aligns with a multivariate Gaussian distribution. Specifically, under a confidence level of 97.5%, the model with our regularization passes the Mardia test in six dimensions, whereas the standard AE fails to do so. These findings are consistent with our visual observations and further validate the effectiveness of the proposed regularization.

| PCA Dim. | Excess Kurt. | Skew. | Z-score | Mardia Test |
|---|---|---|---|---|
| | \multicolumn{3}{c}{(AE $\rightarrow$ AE+GReg)} | | ($\alpha = 0.025$) |
| 2 | $-0.44 \rightarrow$ **0.18** | $0.11 \rightarrow 0.17$ | $-1.76 \rightarrow$ **0.71** | Accept / Accept |
| 5 | $-0.29 \rightarrow$ **-0.12** | $1.32 \rightarrow 1.79$ | $-0.55 \rightarrow$ **-0.22** | Accept / Accept |
| 6 | $1.52 \rightarrow$ **1.30** | $2.24 \rightarrow 3.36$ | $2.45 \rightarrow$ **2.10** | **Reject / Accept** |
| 10 | $8.67 \rightarrow$ **7.15** | $7.03 \rightarrow 8.86$ | $8.85 \rightarrow$ **7.30** | Reject / Reject |
| 50 | $369.55 \rightarrow$ **351.73** | $265.19 \rightarrow$ **261.08** | $81.03 \rightarrow$ **77.12** | Reject / Reject |
| 100 | $1244.82 \rightarrow$ **1173.69** | $1576.51 \rightarrow$ **1537.44** | $137.80 \rightarrow$ **129.93** | Reject / Reject |

Table 1: **Regularization Enhances Gaussian Alignment in Latent Space and PCA Projections.** Statistical comparison between standard autoencoder (AE) and its Gaussian-regularized counterpart (AE+GReg) across varying PCA dimensions. We report excess kurtosis, skewness, and Z-score of the latent distributions, as well as results of the Mardia multivariate normality test at significance level $\alpha = 0.025$. The regularized model consistently achieves lower deviation from Gaussianity, with significant improvements observed in low-dimensional projections (e.g., dimension 6 transitions from rejection to acceptance).

### 4.3 Application to Diverse Generative Models

#### 4.3.1 Autoencoders and Variational Autoencoders

To further validate the impact of our proposed regularization, we compare standard AE and VAE models trained with and without the regularization, under the same training and evaluation settings described in Section 4.2.

**Latent Sampling Strategy for Autoencoder** Unlike VAEs, standard autoencoders do not learn an explicit latent prior, making direct Gaussian sampling infeasible. To enable meaningful generation, we adopt a two-stage sampling procedure for AE models: (1) extract latent vectors from $N$ validation samples using the trained encoder; (2) apply $k$-means clustering to estimate cluster-wise means and covariances ($\boldsymbol{\mu}_i, \boldsymbol{\Sigma}_i$). New latent samples are then drawn from $\mathcal{N}(\boldsymbol{\mu}_i, \boldsymbol{\Sigma}_i)$ and decoded into images. This method preserves the local statistical structure of the latent space while allowing random generation.

**Evaluation Protocol** We evaluate both reconstruction and generation capabilities of each model. For reconstruction, we report Mean Squared Error (MSE), Structural Similarity Index (SSIM), and Learned Perceptual Image Patch Similarity (LPIPS). These metrics jointly assess both pixel-level accuracy and perceptual fidelity. For generation, we compute the Inception Score (IS) and Fréchet Inception Distance (FID). IS captures image quality and class diversity, while FID quantifies the distance between real and generated image distributions in a deep feature space. Among generative metrics, FID is widely regarded as a reliable and sensitive indicator of visual quality and realism.

**Quantitative Results** Table 2 presents the full results. As expected, the standard AE achieves the lowest reconstruction error but performs poorly in generation tasks, primarily due to the lack of probabilistic modeling in its latent space. In contrast, our regularized AE achieves a more favorable trade-off: while its reconstruction error is slightly higher, it attains the best IS and FID scores among the four models, indicating

| Method | Reconstruction | | | Generation | |
|--------|------|------|------|------|------|
| | MSE↓ | SSIM↑ | LPIPS↓ | IS↑ | FID↓ |
| AE | **0.0194** ± 0.0101 | **0.732** ± 0.066 | **0.1442** ± 0.0623 | 1.6837 ± 0.0122 | 59.23 |
| AE+GReg | 0.0216 ± 0.0109 | 0.711 ± 0.069 | 0.1654 ± 0.0636 | **1.9500** ± 0.0379 | **52.42** |
| AE+skew | 0.0242 ± 0.0120 | 0.696 ± 0.070 | 0.1864 ± 0.0695 | 1.8311 ± 0.0210 | 60.94 |
| AE+kurtosis | 0.0324 ± 0.0151 | 0.630 ± 0.079 | 0.2240 ± 0.0792 | 1.7784 ± 0.0271 | 65.43 |
| VAE | 0.0240 ± 0.0104 | 0.696 ± 0.069 | 0.1643 ± 0.0666 | 1.7480 ± 0.0140 | 56.42 |
| VAE+GReg | 0.0240 ± 0.0104 | 0.695 ± 0.069 | 0.1639 ± 0.0662 | 1.7400 ± 0.0160 | 55.19 |
| VQ-VAE-2 | 0.0016 ± 0.0005 | 0.9542 ± 0.0103 | 0.1131 ± 0.0215 | **1.331** ± 0.013 | 72.73 |
| VQ-VAE-2 + GReg | **0.0015** ± 0.0005 | **0.9544** ± 0.0094 | **0.1090** ± 0.0203 | 1.308 ± 0.026 | **69.12** |
| LDM | 0.0049 ± 0.0031 | 0.7458 ± 0.0878 | **0.0782** ± 0.0240 | 2.427 ± 0.057 | 14.15 |
| LDM + GReg | **0.0037** ± 0.0022 | **0.7691** ± 0.0810 | 0.1099 ± 0.0404 | **2.443** ± 0.038 | **11.37** |

Table 2: **Gaussian Regularization Improves Generation Quality Across Encoder-Based Models with Minimal Impact on Reconstruction.** We report quantitative results on both reconstruction and generation performance for AE, VAE, VQ-VAE-2, and LDM models, with and without our proposed Gaussian regularization (GReg). Reconstruction metrics include Mean Squared Error (MSE), Structural Similarity Index (SSIM), and LPIPS; generation quality is assessed using Inception Score (IS) and Fréchet Inception Distance (FID). Regularization leads to consistent gains in generation quality (↑IS, ↓FID) for AE, VQ-VAE-2, and LDM, while maintaining competitive reconstruction fidelity. All experiments are conducted under identical training configurations, and each model is evaluated on a corresponding dataset: CelebA-64 (AE, VAE), RAFD (VQ-VAE-2), and LSUN-Churches (LDM).

significantly improved generative performance. Because the KL divergence already imposes a strong prior on the latent space, leaving limited room for further structural correction, our regularization directly to the VAE yields minimal performance gains. In contrast, AEs benefit more due to their unregularized latent structure.

**Extension to VQ-VAE-2**   To test the effectiveness of our method on discrete latent spaces, we further integrate it into a VQ-VAE-2 model trained on the Radboud Faces Database (RAFD). As shown in Table 2, our method improves generative quality with lower FID, while maintaining good reconstruction performance. These results demonstrate that the proposed regularization is compatible with both continuous and discrete latent representations.

### 4.3.2   Latent Diffusion Models

To further validate the generality of our method in high-resolution generative settings, we integrate the proposed regularization into a Latent Diffusion Model (LDM), which synthesizes images in the latent space of an autoencoder rather than directly in pixel space. To obtain a comparable test-field, we fix the computational resources to two RTX 4090 GPUs for all experiments in this section. LDMs have shown remarkable performance in generating high-quality images with reduced computational cost (Rombach et al., 2022). However, they inherit the independent assumption across dimensions in latent spaces from VAEs, which may significantly degrade generative quality.

We apply our Gaussian regularization term in the latent space during training of the autoencoder stage of LDM, using LSUN-Church-Outdoor dataset. Both model settings are kept identical, including network architecture, learning rate, and training schedule, with the only change being the inclusion of our regularization. After training the first-stage autoencoders, we train diffusion models conditioned on the outputs of the regularized and baseline autoencoders, respectively. This setup allows a fair evaluation of how the latent space distribution affects generation quality.

For both models, we reconstruct all images in the validation set to evaluate the performance of the first-stage autoencoders. We also generate random samples equal in number to the validation set to assess the models' generative capabilities. As shown in Table 2, the regularized LDM encoder achieves better generative

performance, improving both Inception Score (IS) and Fréchet Inception Distance (FID). Although the reconstruction performance slightly degrades, the impact is negligible and well within acceptable margins for diffusion-based models.

Figure 2 presents visual examples of generated samples. Compared to the baseline, our model generates sharper textures and better structural coherence. These improvements can be attributed to the smoother and more Gaussian latent space induced by our regularization.

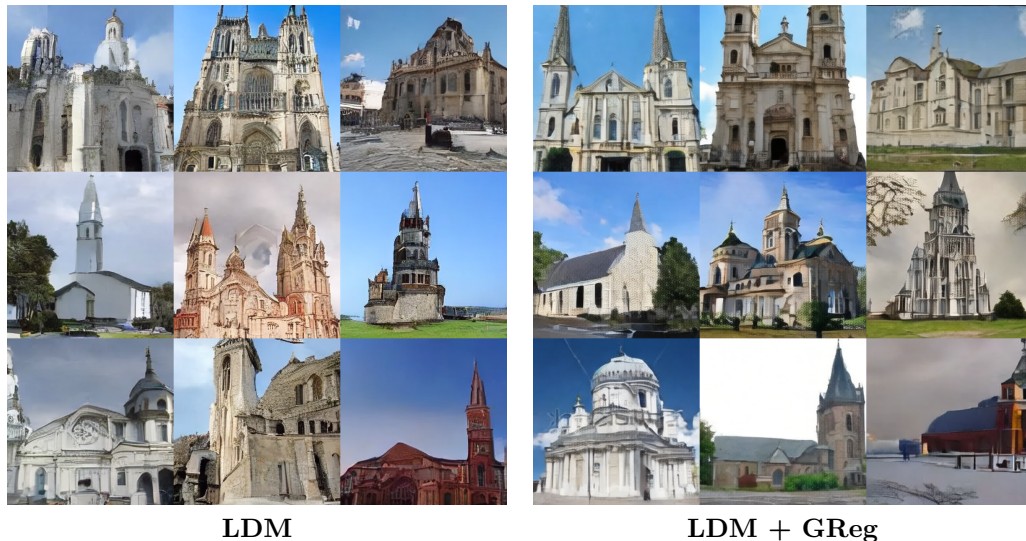

**LDM**                    **LDM + GReg**

Figure 2: **Qualitative comparison of image generation results on the LSUN-Church-Outdoor dataset.** Left: standard Latent Diffusion Model (LDM). Right: LDM with our Gaussian regularization (LDM + GReg). The regularized model generates images with more coherent structures and sharper textures, demonstrating the effectiveness of the proposed regularization in enhancing generative fidelity.

### 4.3.3 StyleGAN2 Inversion and Editing

We further evaluate the effectiveness of our proposed regularization term in the task of StyleGAN inversion and semantic image editing. Specifically, we adopt the StyleGAN2-based e4e framework (Tov et al., 2021), a state-of-the-art encoder designed to project real images into the latent space $\mathcal{W}$, enabling attribute manipulation in linear directions. A central challenge in GAN inversion lies in the trade-off between reconstruction fidelity and editability. Latent codes that yield faithful reconstructions are often far from the native $\mathcal{W}$ manifold, which negatively affects their editability. Conversely, codes closer to $\mathcal{W}$ typically support better semantic editing but may compromise reconstruction quality.

Our regularization addresses this issue by encouraging the latent distribution to approximate a multivariate Gaussian. A key property of Gaussian distributions is closure under affine transformations: for a random vector $\mathbf{z} \sim \mathcal{N}(\boldsymbol{\mu}, \boldsymbol{\Sigma})$, any affine transformation $\mathbf{Az}+\mathbf{b}$ remains Gaussian. This implies that editing operations preserve the statistical structure of the distribution. Consequently, latent codes regularized in this way exhibit greater stability under semantic manipulations, better preserving both realism and identity throughout the editing process.

In practice, we fine-tune the e4e encoder for 5000 steps on the FFHQ dataset with our regularization term applied to its latent space. Evaluation is conducted on CelebA-HQ at a resolution of 1024×1024. As shown in Table 3, our method consistently improves inversion metrics, particularly in LPIPS and FID, indicating enhanced perceptual quality and alignment with the StyleGAN manifold. These results confirm that our regularization effectively reduces distortion while promoting semantic editability.

We further validate this effect through additional editing results. As shown in Figure 3, our regularized model (e4e + GReg) is able to complete missing facial regions, particularly in the hair area, when given incomplete

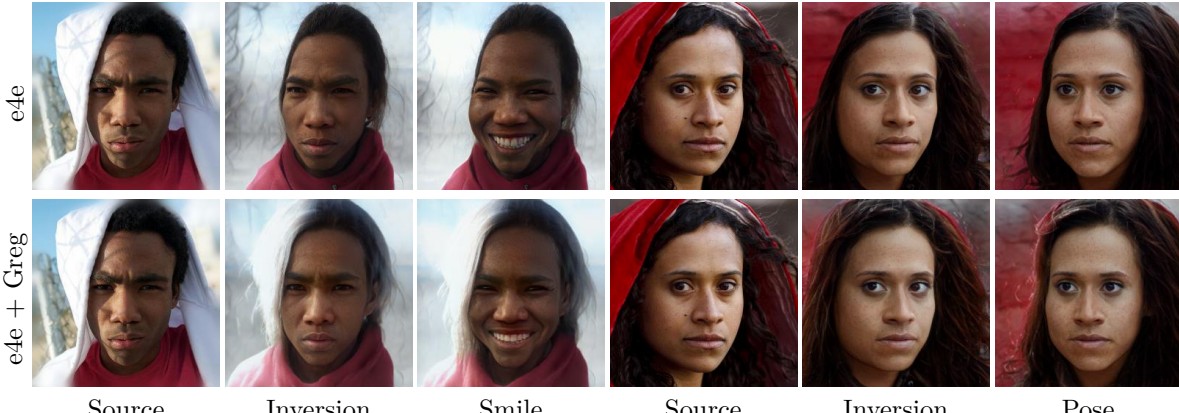

Figure 3: **Visual comparison of inversion and smile editing results using the original e4e model (top) and the Gaussian-regularized version (bottom) on incomplete face inputs.** The e4e model with our Gaussian regularization automatically completes missing facial regions, particularly in the hair area, while preserving alignment with the facial manifold. This consistency is maintained throughout subsequent editing operations.

| Method | MSE↓ | SSIM↑ | LPIPS↓ | FID↓ |
|---|---|---|---|---|
| e4e | $0.0150 \pm 0.0112$ | $0.5098 \pm 0.0937$ | $0.2250 \pm 0.0656$ | 46.59 |
| e4e + GReg | $\mathbf{0.0143} \pm 0.0111$ | $\mathbf{0.5138} \pm 0.0952$ | $\mathbf{0.2218} \pm 0.0670$ | **45.98** |

Table 3: **Quantitative comparison of inversion results from the original e4e model and its Gaussian-regularized counterpart (e4e + GReg)**. The proposed e4e + GReg consistently outperforms the baseline across all pixel-level metrics. Notably, improvements in LPIPS and FID indicate enhanced perceptual quality and better alignment with the StyleGAN2 latent manifold, leading to improved editability and semantic consistency during attribute manipulation.

face image inputs. This not only preserves alignment with the facial manifold during inversion but also maintains consistency under subsequent semantic edits. In addition, Figure 4 presents editing comparisons using InterfaceGAN (Shen et al., 2020). Our method better preserves fine-grained facial details (e.g., eyes, mouth, hair) and produces smoother transitions when modifying attributes. This demonstrates that our approach not only improves latent alignment, but also enables more faithful and expressive manipulations within the GAN latent space.

## 4.4 User Study

To increase the credibility of qualitative results, we conduct a user study to evaluate perceptual quality of outputs produced by baseline models and their Gaussian-regularized counterparts.

For e4e, the respondents are given a side-by-side comparison of images reconstructed and edited (smile) by e4e and its Gaussian-regularized counterpart, and are asked to choose the more realistic image. For e4e, the shown images are of results obtained after performing editing on inverted latent codes (smile). For LDM, the images shown are changed to unconditional generation results. The results of the user study are shown in Table 4. As can be seen, the perceptual quality of images obtained by e4e+GReg is higher both on the reconstructed and edited images and the generation results of LDM+GReg are also better under statistical hypothesis test. For e4e, GReg consistently enhances both reconstruction fidelity and attribute-editing accuracy, suggesting that a more Gaussian latent space aids both pixel-level and semantic control tasks. For LDM, despite its already high generation quality, GReg still provides a statistically significant perceptual benefit, indicating that distributional regularization can complement powerful generative priors.

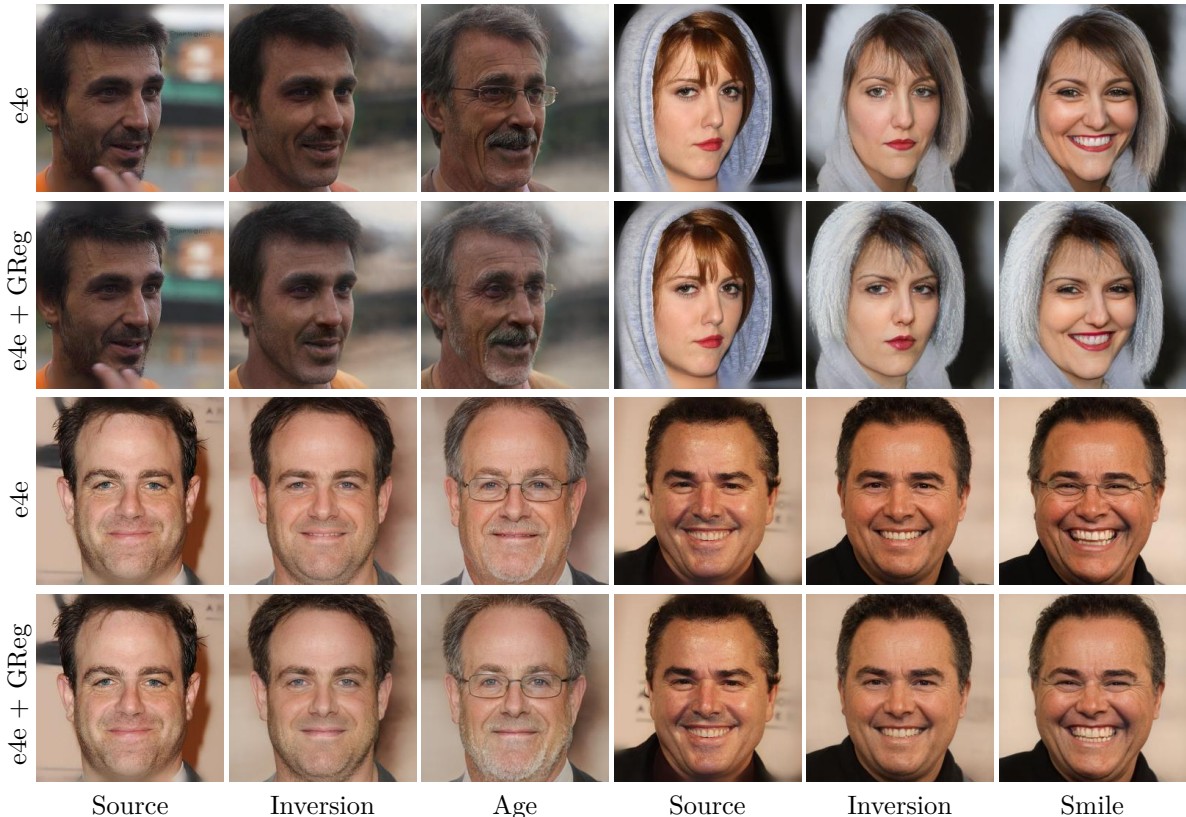

Figure 4: **Gaussian Regularization Enhances Inversion Quality and Editing Consistency in Style-GAN2 Inversion.** Inversion and attribute editing results using the original e4e model (odd rows) and its Gaussian-regularized counterpart (even rows). The regularized model produces reconstructions that better preserve identity and structure, and maintains fine-grained details such as eyes, hair, and facial contours under strong attribute changes. These improvements highlight the benefit of aligning latent codes with a Gaussian prior in enhancing semantic consistency and editability.

| Task / Model | Baseline (%) | GReg (%) | $p$-value |
|---|---|---|---|
| e4e Reconstruction | 37.01 | **62.99** | $<0.001$ |
| e4e Smile Editing | 37.47 | **62.52** | $<0.01$ |
| LDM Generation | 33.21 | **66.79** | $1.49 \times 10^{-22}$ |

Table 4: Aggregated human evaluation results: proportion of votes for baseline and GReg (excluding "Same"), and the corresponding two-sided binomial test $p$-value.

### 4.5 Robustness Analysis

We evaluate the robustness of our proposed Gaussian regularization under varying training and generation settings using an autoencoder. Specifically, we investigate the effects of three factors: the regularization strength $\lambda$, the training batch size, and the different initialization for generation.

#### 4.5.1 Effect of Regularization Strength $\lambda$

We assess $\lambda$ values in $\{0.001, 0.002, 0.005, 0.01, 0.05, 0.1, 0.2, 0.5, 1, 2, 5, 10\}$ using standard reconstruction and generation metrics. As shown in Table 6 and Fig.5, very small regularization strengths ($\lambda \leq 0.05$) yield slightly better reconstruction fidelity, but noticeably lower generation quality. Increasing $\lambda$ up to approximately 0.1 improves IS and reduces FID, with the best trade-off observed in the range $\lambda \in [0.1, 0.5]$. For

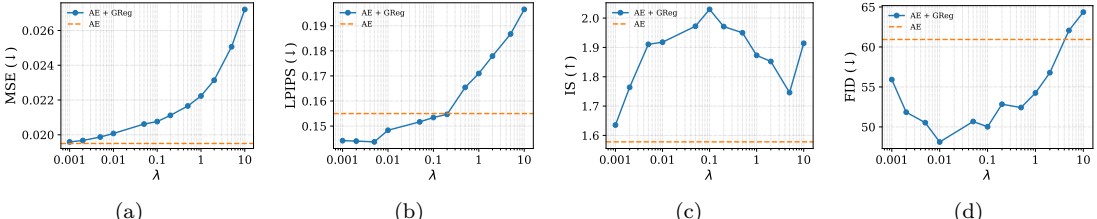

Figure 5: **Effect of regularization strength $\lambda$ on reconstruction and generation performance of AE+GReg.** (a) MSE, (b) LPIPS, (c) Inception Score (IS), and (d) Fréchet Inception Distance (FID) are reported for $\lambda \in \{0.001, 0.002, 0.005, 0.01, 0.05, 0.1, 0.2, 0.5, 1, 2, 5, 10\}$. Small $\lambda$ values favor reconstruction but yield lower generative quality, while moderate values ($0.1 \leq \lambda \leq 0.5$) achieve the best overall trade-off. Excessively large $\lambda$ leads to performance degradation in both tasks.

excessively large values ($\lambda \geq 5$), both reconstruction and generation performance degrade due to over-regularization.

### 4.5.2 Effect of Batch Size

We investigate the impact of training batch size on the performance of AE+GReg by varying the batch size in $\{8, 16, 32, 64, 128\}$ while keeping all other training configurations fixed. Table 7 reports reconstruction and generation metrics. According to Figure 8 (a) and (b), IS varies only by $\approx 9.1\%$ and FID within $\approx 2.6$, indicating that batch size has little influence on generative performance. Reconstruction metrics also remain stable, with only minor fluctuations in MSE, SSIM and LPIPS.

### 4.5.3 Effect of Generation Initialization

We further examine the effect of random initialization on generation quality by evaluating AE+GReg with 10 different seeds. The trained model remains fixed across trials ($\lambda = 0.5$). As shown in Table 8 (c) and (d), the IS varies by only $\approx 9.1\%$ and FID within $\approx 2.65$, confirming that the choice of generation initialization has negligible impact on generative performance.

## 5 Conclusions

In this work, we propose a general and theoretically grounded regularization method for encoder-based models, encouraging latent vectors to align with a multivariate Gaussian prior. By employing random projections, our method offers a practical approach to assess high-dimensional Gaussian properties without explicit density estimation. Notably, in contrast to standard variational autoencoder structures that assume independent components in the latent space, our method relaxes this assumption and supports general multivariate Gaussian priors. This flexibility allows the model to capture dependencies inherent in real-world data, resulting in more expressive and semantically meaningful latent representations. Moreover, the proposed regularization introduces no additional computational overhead and can be seamlessly integrated into the training objectives of various encoder-based models. Extensive experiments across diverse architectures demonstrate that our method consistently improves generation quality while preserving high reconstruction fidelity. Its compatibility with discrete latent spaces and effectiveness in enhancing semantic editability in StyleGAN-based frameworks are also verified. Overall, our approach provides a simple yet effective regularization strategy for shaping latent spaces in encoder-based generative models. It lays a foundation for further exploration of both the theoretical properties and probabilistic modeling of latent structures. This method has the potential to benefit a wide range of applications, including style transfer, conditional synthesis, semantic editing, and other tasks requiring Gaussian latent representations. However, while our method shows broad applicability across encoder-based models, it currently assumes a latent space with finite moments and may exhibit sensitivity to batch statistics in extreme scenarios. Addressing these limitations through more robust formulations is a promising direction for future research.

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

## A  Proof of Theorem 1

**Lemma 2.** *(Bentkus, 2005)*

*Let $\mathbf{X}_1, \ldots, \mathbf{X}_n$ be independent random vectors in $\mathbb{R}^d$ with zero mean. Denote $S = \sum_{i=1}^{n} X_i$, and assume the covariance matrix $\mathbf{C}^2$ of $\mathbf{S}$ is invertible. Let $\mathbf{W}$ be a zero-mean Gaussian vector with the same covariance as $\mathbf{S}$. For the collection $\mathcal{C}$ of all convex Borel sets in $\mathbb{R}^d$, the following Lyapunov-type inequality holds:*

$$\sup_{A \in \mathcal{C}} |P\{\mathbf{S} \in A\} - P\{\mathbf{W} \in A\}| \le c d^{\frac{1}{4}} \sum_{k=1}^{n} \mathbb{E} \|\mathbf{C}^{-1} \mathbf{X}_k\|_2^3 \tag{9}$$

*where $c$ is a positive constant.*

*Proof of Theorem 1.* From the definition of sparse random projections, each projected latent vector can be expressed as:

$$\mathbf{z}_i' = \frac{1}{\sqrt{l}} \mathbf{R}^T \mathbf{z}_i, \qquad i = 1, \cdots, n \tag{10}$$

This projection can be decomposed into a sum of independent random vectors:

$$\mathbf{z}_i' = \frac{1}{\sqrt{l}} \sum_{j=1}^{m} \begin{pmatrix} r_{j1} z_{i,j} \\ r_{j2} z_{i,j} \\ \vdots \\ r_{jl} z_{i,j} \end{pmatrix} = \sum_{j=1}^{m} \boldsymbol{u}_j \tag{11}$$

As a result, the covariance matrix of $\mathbf{z}_i'$ can be written as the sum of covariances of the $\boldsymbol{u}_j$:

$$Cov(\mathbf{z}_i) = Cov\left( \sum_{j=1}^{m} \boldsymbol{u}_j \right) = \sum_{j=1}^{m} Cov(\boldsymbol{u}_j) = \frac{1}{l} \left( \sum_{j=1}^{m} z_i j^2 \right) \boldsymbol{I}_l \tag{12}$$

Applying Lemma 2, we obtain:

$$|P\{\mathbf{z}_i' \in B\} - P\{\boldsymbol{w}_i \in B\}|$$

$$\le C(m) \sum_{j=1}^{m} \left\| \left( \frac{1}{l} \left( \sum_{j=1}^{m} z_i j^2 \right) \boldsymbol{I}_l \right)^{-\frac{1}{2}} \boldsymbol{u}_j \right\|_2^3$$

$$= \frac{C(m)}{\left( \sum_{j=1}^{m} z_{ij}^2 \right)^{\frac{3}{2}}} \sum_{j=1}^{m} \left\| \begin{pmatrix} r_{j1} z_{i,j} \\ r_{j2} z_{i,j} \\ \vdots \\ r_{jl} z_{i,j} \end{pmatrix} \right\|_2^3 \tag{13}$$

$$= \frac{C(m)}{\left( \sum_{j=1}^{m} z_{ij}^2 \right)^{\frac{3}{2}}} \sum_{j=1}^{m} \mathbb{E} \left[ \sum_{k=1}^{l} r_{jk}^2 \right]^{\frac{3}{2}} |z_{i,j}|^3$$

$$= C(m) e(l) \frac{\sum_{j=1}^{m} |z_{i,j}|^3}{\left( \sum_{j=1}^{m} z_{ij}^2 \right)^{\frac{3}{2}}}$$

where $e(l) = \mathbb{E} \left[ \sum_{k=1}^{l} r_{jk}^2 \right]^{3/2}$ is a constant depending only on the projection dimension $l$. Recent work has shown that $C(m)$ can be upper-bounded by $42m^{1/4} + 16$ under mild assumptions (Raič, 2019).

$\square$

# B  Detailed Experimental Settings

**Experiments on AE and VAE**  We implement the Autoencoder (AE) model following the network architecture in Hou et al. (2017), with a latent dimension of 100. The encoder comprises four convolutional layers with increasing channels [32, 64, 128, 256], each followed by BatchNorm and LeakyReLU activations. The decoder mirrors this structure using transposed convolutions and upsampling operations. All models are trained on the CelebA-64 dataset for 40 epochs using the Adam optimizer with an initial learning rate of 0.0005 and a batch size of 64. The learning rate is halved every 5 epochs via a step scheduler (`gamma=0.5`). Training is conducted on a single NVIDIA RTX 4090D GPU. The reconstruction loss is computed using mean squared error. To investigate the structure of the learned latent space and the quality of generated samples, we perform generation experiments using the trained AE model, following the method introduced in Section 4.3.1. Specifically, we randomly select 1,000 samples from the CelebA-64 training dataset and encode them into latent features. These features are then clustered into 50 groups using k-Means. For each cluster, we compute the empirical mean and covariance of the encoded latent vectors.

**Experiments on LDMs**  We adopt a two-stage training for Latent Diffusion Models (LDM) on the LSUN-Church-Outdoor dataset, following the standard LDM. We fix the computational resources to two RTX 4090 GPUs. Both stages are trained from scratch under a fixed resolution of 256×256.

In the first stage, we train a KL-regularized autoencoder with latent dimension 4 to learn a compact and expressive latent space for natural images. The encoder-decoder structure uses 4 downsampling blocks with channel multipliers [1,2,4,4] and 2 residual blocks per stage. The decoder mirrors this structure. KL divergence is softly weighted ($10^{-6}$) to maintain high visual fidelity. We use a warmup of 50,000 steps, adversarial weight 0.5, and penultimate feature loss weight 1.0. The base learning rate is set to 4.5e-6, optimized using gradient accumulation over 6 steps with a batch size of 4. After training, we reconstruct all images in the validation set to evaluate the first-stage autoencoder's reconstruction quality. For regularized LDM, we impose the regularization in the last layer of latent space, because linear transformation will not affect Gaussian properties. We set the regularization weight to 1.

In the second stage, we train a DDPM-style Latent Diffusion Model (LDM) in the learned latent space of dimension 4, where all settings follow those in Rombach et al. (2022). The diffusion model operates over a 32×32 latent spatial grid with 4 channels. The model is trained for 1000 diffusion steps, with the noise schedule linearly interpolated between 0.0015 and 0.0155. We use L1 loss for simplicity and stability. The learning rate follows a custom warm-up and linear decay scheduler with 10,000 warmup steps, and a base learning rate of 5.0e-5. The model is trained and sampled in an unconditional setting. The pretrained autoencoder from Stage 1 is frozen.

| Dataset | LSUN-Church-Outdoor | Resolution | 256×256 |
|---|---|---|---|
| Latent Dim | 4 | Ch. Mult | [1, 2, 4, 4] |
| Base LR | 4.5e-6 | Batch Size | 4 |
| Grad Accum | 6 | Optimizer | Adam |
| **KL Weight** | 1e-6 | **GReg Weight** | 1 |
| Loss Func | Fidelity + **KL or Our Gaussian Regularization** | | |

Table 5: Autoencoder (Stage 1) Training Configuration. Settings are used for both baseline and regularized models to ensure a fair comparison.

**Experiments on e4e**  We follow the original e4e framework structure as proposed in Tov et al. (2021), keeping all architectural components unchanged. The model is trained on the FFHQ dataset at a resolution of 1024×1024 and evaluated on the CelebA-HQ dataset, using the official pretrained StyleGAN2 weights as the generator. All hyperparameters remain consistent with those in Tov et al. (2021). We fix the computational resources to two RTX 4090 GPUs. We fine-tune the e4e encoder for 5,000 steps by incorporating the proposed Gaussian regularization into the encoder's latent space, using a regularization weight of 1.0.

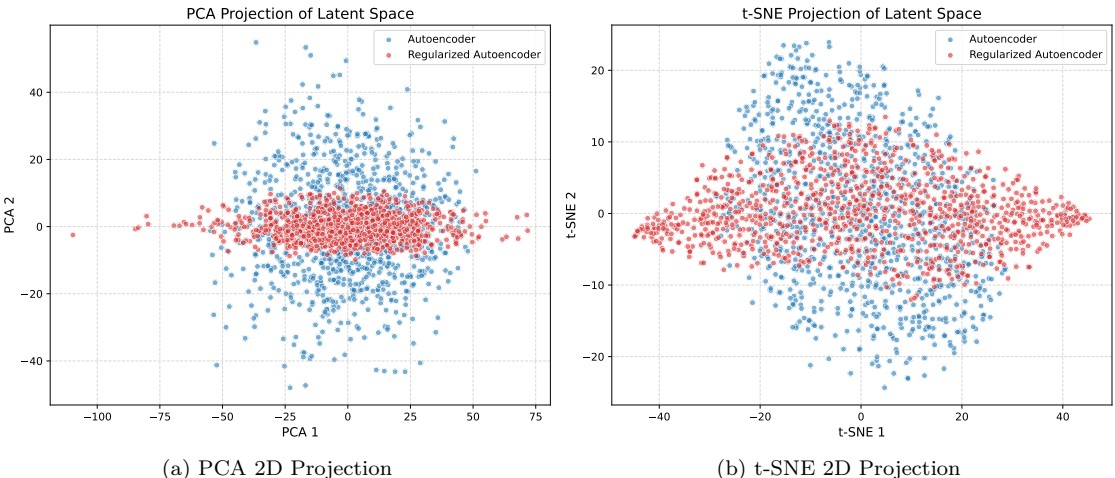

(a) PCA 2D Projection        (b) t-SNE 2D Projection

Figure 6: **Regularization Encourages Gaussian Structure in Latent Space: PCA and t-SNE Visualizations.** 2D projections of latent vectors from a standard autoencoder (blue) and its Gaussian-regularized counterpart (red). Left (PCA): The regularized model yields a compact, elliptical shape—typical of multivariate Gaussians—indicating improved global alignment. Right (t-SNE): The regularized latent space exhibits tighter local clusters, indicating enhanced consistency and structure in learned representations.

## C   Projection Results

We show the projection results for Section 4.2 in Figure 6.

## D   Supplement for the effect of Regularization Strength $\lambda$

As the supplement of Section 4.5.1, we present detailed results for different regularization strengths, as shown in Table 6.

| $\lambda$ | MSE | PSNR | LPIPS | IS | FID |
|---|---|---|---|---|---|
| 0.001 | $0.019591 \pm 0.010197$ | $23.5462 \pm 2.0482$ | $0.144192 \pm 0.062392$ | $1.6354 \pm 0.0171$ | 55.9205 |
| 0.002 | $0.019671 \pm 0.010257$ | $23.5277 \pm 2.0497$ | $0.144025 \pm 0.062211$ | $1.7639 \pm 0.0185$ | 51.8375 |
| 0.005 | $0.019872 \pm 0.010352$ | $23.4827 \pm 2.0482$ | $0.143712 \pm 0.061855$ | $1.9106 \pm 0.0186$ | 50.5274 |
| 0.01 | $0.020075 \pm 0.010429$ | $23.4353 \pm 2.0412$ | $0.148345 \pm 0.062514$ | $1.9178 \pm 0.0178$ | 48.1115 |
| 0.05 | $0.020617 \pm 0.010636$ | $23.3118 \pm 2.0257$ | $0.151684 \pm 0.062651$ | $1.9722 \pm 0.0144$ | 50.6776 |
| 0.1 | $0.020760 \pm 0.010700$ | $23.2816 \pm 2.0260$ | $0.153427 \pm 0.062981$ | $2.0297 \pm 0.0372$ | 50.0153 |
| 0.2 | $0.021117 \pm 0.010807$ | $23.2022 \pm 2.0139$ | $0.154684 \pm 0.063166$ | $1.9711 \pm 0.0259$ | 52.8373 |
| 0.5 | $0.021657 \pm 0.010929$ | $23.0802 \pm 1.9858$ | $0.165414 \pm 0.063592$ | $1.9500 \pm 0.0379$ | 52.4236 |
| 1 | $0.022232 \pm 0.011055$ | $22.9560 \pm 1.9574$ | $0.170973 \pm 0.064062$ | $1.8729 \pm 0.0252$ | 54.2465 |
| 2 | $0.023140 \pm 0.011273$ | $22.7650 \pm 1.9186$ | $0.177906 \pm 0.065701$ | $1.8522 \pm 0.0194$ | 56.7838 |
| 5 | $0.025064 \pm 0.011874$ | $22.3868 \pm 1.8624$ | $0.186704 \pm 0.066650$ | $1.7461 \pm 0.0198$ | 62.0605 |
| 10 | $0.027211 \pm 0.011885$ | $21.9698 \pm 1.7209$ | $0.196515 \pm 0.066763$ | $1.9140 \pm 0.0141$ | 64.3632 |

Table 6: **Detailed results of reconstruction and generation performance for different regularization strengths**

# E    Supplement for the effect of Batch Size

As the supplement of Section 4.5.2,we present detailed generation results for different batch sizes, as shown in Table 7 and Figure 8. And we present the reconstruction results in Figure 7.

| Batch | MSE↓ | SSIM↑ | LPIPS↓ | IS↑ | FID↓ |
|---|---|---|---|---|---|
| 8 | 0.021284 ± 0.010847 | 0.7149 ± 0.0697 | 0.1550 ± 0.0633 | 1.7879 ± 0.0162 | 52.47 |
| 16 | 0.021166 ± 0.010803 | 0.7156 ± 0.0696 | 0.1575 ± 0.0630 | 1.8879 ± 0.0255 | 52.75 |
| 32 | 0.021342 ± 0.010809 | 0.7139 ± 0.0694 | 0.1616 ± 0.0632 | 1.9408 ± 0.0279 | 51.42 |
| 64 | 0.021657 ± 0.010929 | 0.7113 ± 0.0693 | 0.1654 ± 0.0636 | 1.9500 ± 0.0379 | 52.42 |
| 128 | 0.022178 ± 0.011094 | 0.7051 ± 0.0690 | 0.1682 ± 0.0634 | 1.8913 ± 0.0262 | 54.01 |

Table 7: **Effect of training batch size on AE+GReg performance.**

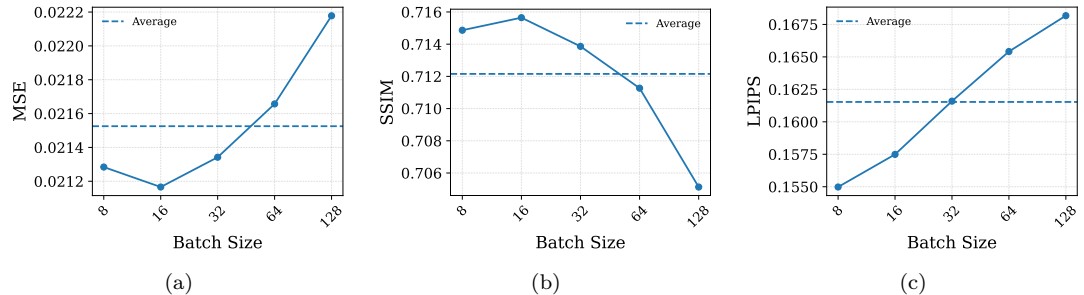

(a)          (b)          (c)

Figure 7:  **Effect of batch size on reconstruction performance of AE+GReg.** (a) MSE, (b) SSIM and (c) LPIPS, are reported for batch size$\in \{8, 16, 32, 64, 128\}$.

# F    Supplement for the effect of Different Generation Initialization

As the supplement of Section 4.5.3, we present detailed results for 10 different sampling seeds, as shown in Table 8 and Figure 8.

| Seed | IS | FID |
|---|---|---|
| 42 | 1.9100 ± 0.0194 | 51.9498 |
| 123 | 1.9187 ± 0.0263 | 51.4774 |
| 456 | 1.8851 ± 0.0318 | 52.1699 |
| 789 | 1.8951 ± 0.0255 | 52.4961 |
| 101112 | 2.0112 ± 0.0262 | 52.2983 |
| 131415 | 1.8483 ± 0.0182 | 49.6757 |
| 161718 | 1.9377 ± 0.0209 | 51.9644 |
| 192021 | 1.8436 ± 0.0187 | 51.1513 |
| 222324 | 2.0102 ± 0.0309 | 52.3296 |
| 252627 | 1.9514 ± 0.0320 | 51.9782 |

Table 8: **Effect of sampling initialization on AE+GReg generation performance.**

# G    More Qualitative Results for StyleGAN Inversion and Editing

As the supplement of Section 4.3.3, we present more qualitative results in Figure 9 and 10 for e4e (StyleGAN Inversion and Editing) and its Gaussian-regularized counterpart.

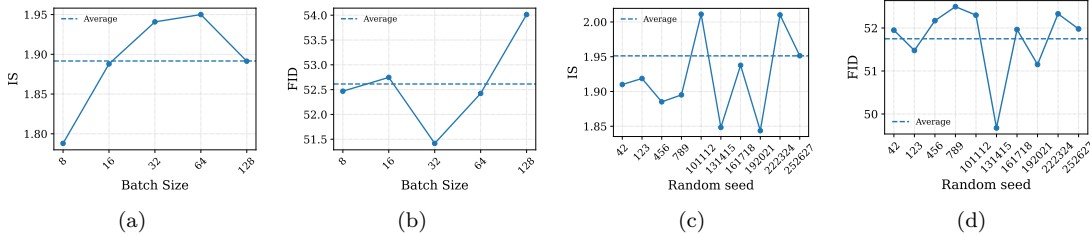

(a)  (b)  (c)  (d)

Figure 8: **Effect of batch size and initialization on generation performance of AE+GReg.** (a) IS for batch sizes, (b) FID for batch sizes, (c) IS for seeds, and (d) FID for seeds are reported for batch size$\in \{8, 16, 32, 64, 128\}$ and 10 different random seeds. For batch size experiments, IS varies by $\approx 9.1\%$ and FID within $\approx 2.6$. For initialization experiments, IS varies by $\approx 9.1\%$ and FID within $\approx 2.65$.

## H  Detailed Data for User Studies

As the supplement of Section 4.4, we present the detailed data for user studies in Table 9 and 10.

| Sample Number | Reconstruction | | | Editing | | |
|---|---|---|---|---|---|---|
| | e4e | e4e+GReg | same | e4e | e4e+GReg | Same |
| 1 | 19 | 81 | 0 | 34 | 66 | 0 |
| 2 | 33 | 67 | 0 | 38 | 62 | 0 |
| 3 | 40 | 60 | 0 | 32 | 68 | 0 |
| 4 | 39 | 61 | 0 | 28 | 72 | 0 |
| 5 | 34 | 66 | 0 | 42 | 58 | 0 |
| 6 | 27 | 62 | 11 | 50 | 45 | 5 |
| 7 | 35 | 59 | 6 | 39 | 56 | 5 |
| 8 | 42 | 54 | 4 | 18 | 77 | 5 |
| 9 | 30 | 62 | 8 | 42 | 48 | 10 |
| 10 | 57 | 34 | 9 | 39 | 52 | 9 |

Table 9: User Study of e4e and e4e+GReg performance on reconstruction and editing, with a total of 100 users.

| Sample Number | Generation | | |
|---|---|---|---|
| | LDM | LDM+GReg | Same |
| 1 | 19 | 77 | 4 |
| 2 | 19 | 70 | 11 |
| 3 | 25 | 53 | 22 |
| 4 | 22 | 46 | 32 |
| 5 | 32 | 61 | 7 |
| 6 | 25 | 34 | 41 |
| 7 | 44 | 46 | 10 |
| 8 | 36 | 47 | 17 |
| 9 | 21 | 72 | 7 |
| 10 | 35 | 53 | 12 |

Table 10: User Study of LDM and LDM+GReg performance on the generation, with a total of 100 users.

## I  E4e Editing Metrics

As the supplement of Section 4.3.3, we present the line plots for image editing (smile, age, pose) metrics in Table 11, including LPIPS, IS and FID.

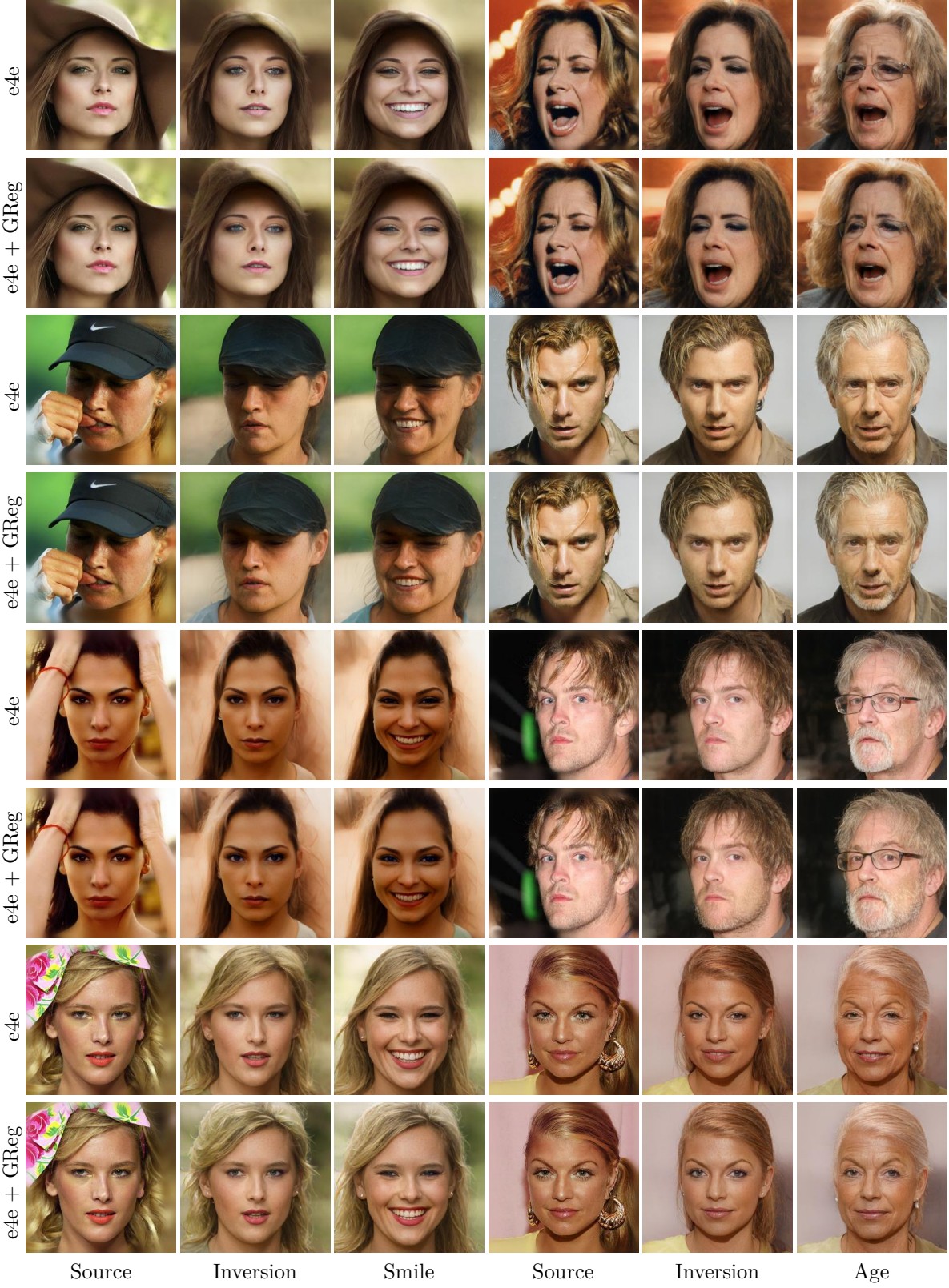

Figure 9: **Gaussian Regularization Enhances Inversion Quality and Editing Consistency in Style-GAN2 Inversion.** Inversion and attribute editing results using the original e4e model (odd rows) and its Gaussian-regularized counterpart (even rows).

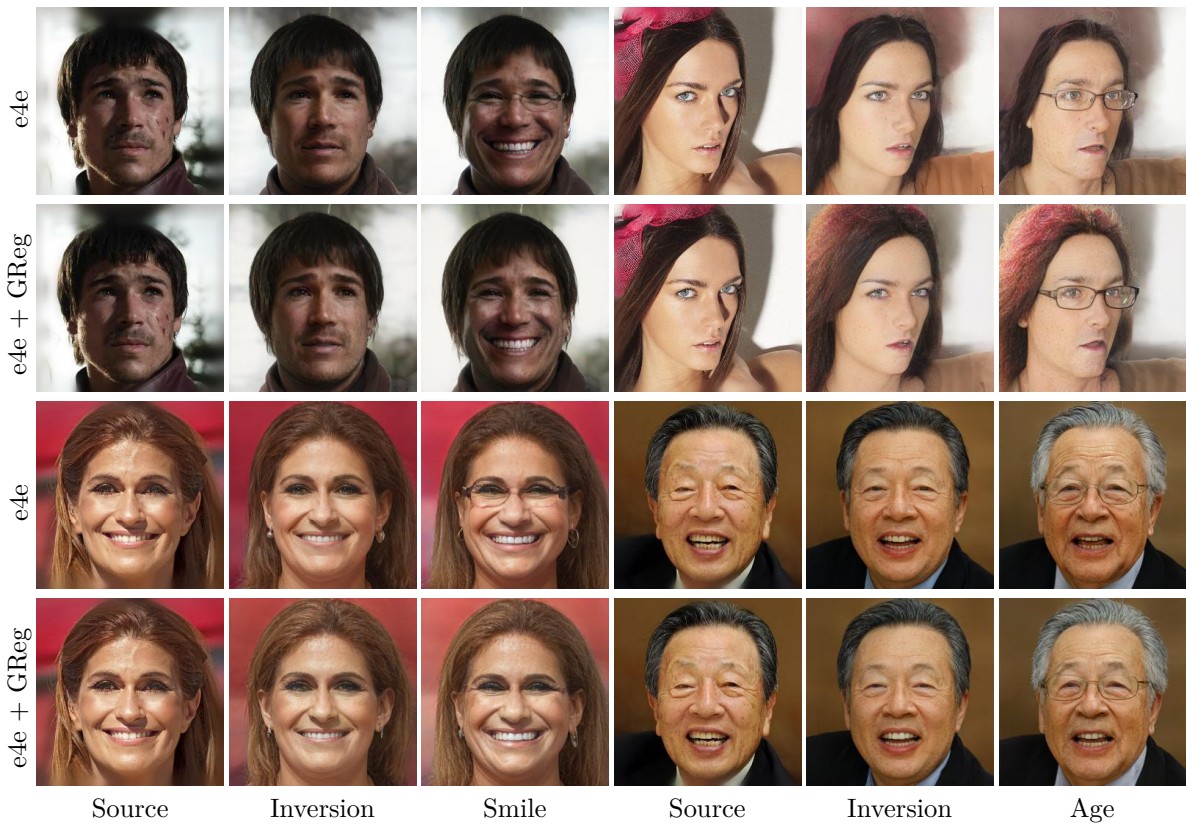

Figure 10: **Gaussian Regularization Enhances Inversion Quality and Editing Consistency in StyleGAN2 Inversion.** Inversion and attribute editing results using the original e4e model (odd rows) and its Gaussian-regularized counterpart (even rows).

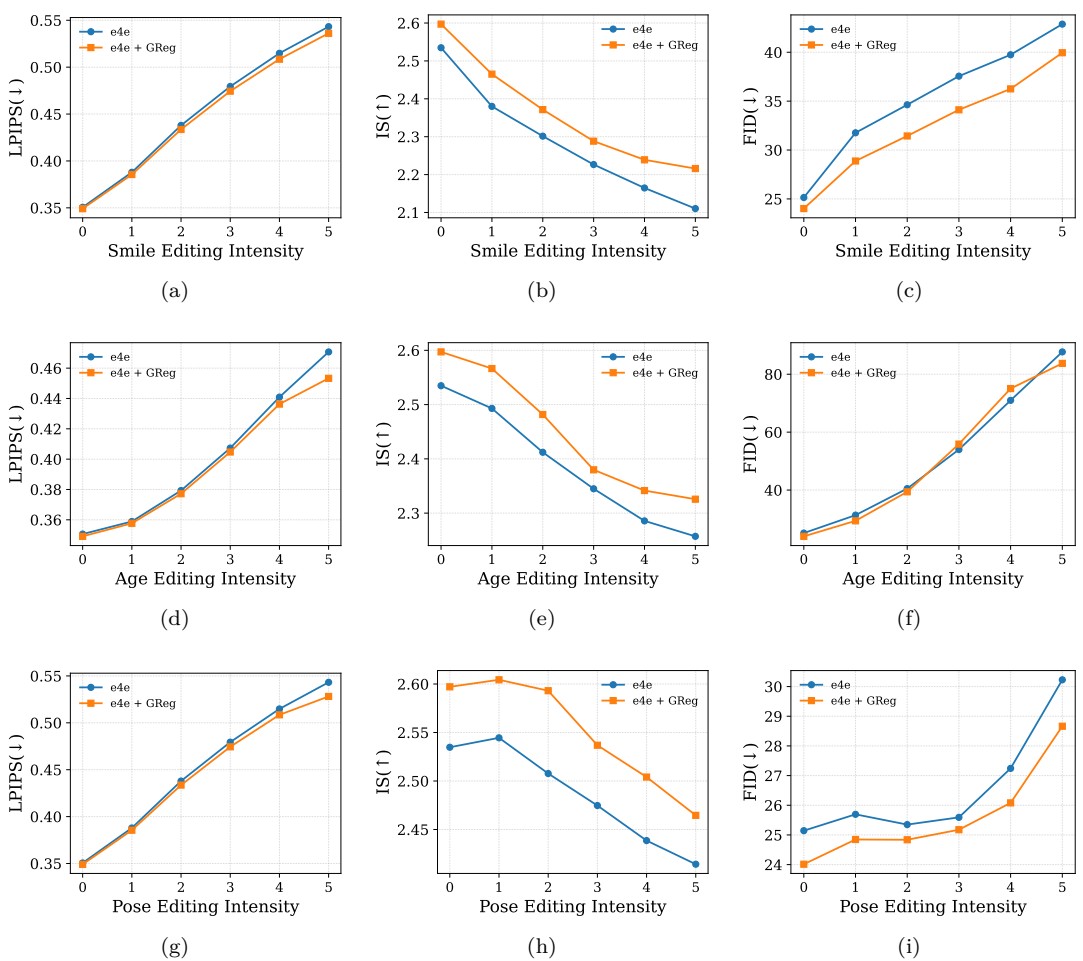

Figure 11: **LPIPS, IS and FID for different editing (smile, age, pose) intensity.**

