# OpenReview forum: "A General Constraint for Gaussian Latent Variables"
_TMLR — Rejected by TMLR_

### Review · Reviewer_jUgh · 2025-07-29

**Summary Of Contributions:**

In this paper, the authors propose a regularized autoencoder type method that can replace the Gaussian regularizer we normally use in the VAE. The method does not minimize ELBO as in the VAE - which results in minimizing the KL divergence between the prior and encoded outputs. In the current method, they make the encoded embeddings similar to a gaussian by deriving a third order skewness type term in the regularizer. This third order moment is shown to penalize deviation from gaussian through a 'Berry-Essen' type bound, wherein they show that lower dimensional projections satisfy the bound produced - and hence the bound can be used as a regularizer.

Main contributions:
- A general third order regularizer that makes the latent variables more gaussian-like
- Some computational savings (since it doesn't need a variance arm)
- Evaluations effective in a number of models (VAEs, StyleGAN, LDMs, ...) showing efficacy and general usability of regularizer term

**Additional Comments:**

This is a nice paper. I think we should just add a bit more material to make the paper a bit easier to read. See my comments above.

**Audience:**

Yes

**Audience Explanation:**

This work is of interested to the machine learning community generally, building upon generative modelling works.

**Claims And Evidence:**

Yes

**Claims Explanation:**

The claims seem to bode well, with one little issue. I think I need more evidence (mathematical, and graphical) how the random projection satisfies the bound. I don't think I understand the ternary matrix. Can the authors also show visual evidence that this indeed works, through  toy examples?

**Requested Changes:**

Please add more clarification on the ternary matrix R, with more examples for clarity. Likewise, please also add more visuals if possible showing evidence that the term makes the latents more gaussian. Likewise, I would also appreciate more mathematical exposition for clarity.

---

> ### Author Response · Authors · 2025-08-15
> **Response to Reviewer jUgh**
>
> Thank you for your comprehensive and in-depth review. We have submitted a revised version of our manuscript, in which all modifications to the main text are highlighted in blue.
>
> **Q1：Explanation of the ternary matrix**
>
> **A1:**
> Throughout the paper, we use a ternary random matrix whose entries are i.i.d. and take values in {-1, 0,1}. Each entry equals 1 with probability $\dfrac{1}{2p}$ , -1 with probability $\dfrac{1}{2p}$ and 0 with probability $1 - \dfrac{1}{p}$. In the paper, we choose an $m \times l$ matrix, which enables the random projection of an $m$-dimensional random vector onto an $l$-dimensional space.
>
> **Q2：Please also add more visuals if possible showing evidence that the term makes the latents more gaussian.**
>
> **A2:**
> We add a random-projection experiment on synthetic Gamma data (projecting from $100$ dimensions to $1$). We generate data matrices by sampling from $\mathrm{Gamma}(\alpha,\text{rate}=1)$ while varying the shape parameter $\alpha \in \{0.02, 0.05, 0.08, 0.1, 0.2, 0.5, 1, 5, 10\}$. Results appear in **Section 4.1 and Figure 1**. As the original data’s $L_{\mathrm{GReg}}$ decreases, the projected distribution becomes increasingly Gaussian.

---

> > ### Comment · Reviewer_jUgh · 2025-08-28
> > **Author Responses**
> >
> > Thanks for the insightful responses. I stick to my original impression that this method of adding a regularizer is interesting and potentially useful without resorting to things like adversarial optimization (like in adversarial autoencoders). The VQ-VAE part seems like a distraction - does not add to the main story. But aside from that I like this work.

---

### Review · Reviewer_t67c · 2025-08-03

**Summary Of Contributions:**

The paper considers autoencoders and proposes to add a regularization term based on the normalized third moment.  For VAEs, this regularization term replaces the KL divergence.
The goal is to regularize the latent representations toward more Gaussian solutions.

The authors consider a range of experiments, particularly image generation, showing their method sometimes improves metrics.

**Audience:**

No

**Audience Explanation:**

The authors propose a simple modification to the training objective of AEs and VAEs and I am not yet convinced by the evidence.

**Claims And Evidence:**

No

**Claims Explanation:**

Often the paper does not report error bars on their metrics, making it difficult to ascertain the significance of the results.

On the Mardia test, there is no adjustment for multiple testing.

Regarding the sampling from AE latent spaces -the authors use k-means + sampling. This can be susceptible to poor initializations - can the authors investigate more how this affects results for both AE and AE+GR?

Regarding diffusions - the diffusions are trained based on the fixed latent spaces learned by AE or AE+GR. Are the metrics computed over many different initializations and trainings of the AEs and AE+GR?

**Requested Changes:**

- I am not convinced the latent projection theory is necessary. The motivation is that Gaussianity is hard to test in high dimensions, so latent projections are warranted. But the resulting regularization term does not involve latent projections - it is just the normalized third moment, which should be close to zero for a Gaussian anyway. Consequently, I think the theoretical contribution should be minimized.

- Often the paper does not report error bars on their metrics, making it difficult to ascertain the significance of the results.

- On the Mardia test, there is no adjustment for multiple testing.

- Regarding the sampling from AE latent spaces -the authors use k-means + sampling. This can be susceptible to poor initializations - can the authors investigate more how this affects results for both AE and AE+GR?

- Regarding diffusions - the diffusions are trained based on the fixed latent spaces learned by AE or AE+GR. Are the metrics computed over many different initializations and trainings of the AEs and AE+GR?

- I am confused how Gaussian regularization helps VQ-VAEs. VQ-VAEs have discrete latent spaces, not continuous. Can the authors explain?

- how is lambda selected?

- can the authors add more experiments on how sensitive the method is to batch effects?

---

> ### Author Response · Authors · 2025-08-15
> **Response to Reviewer t67c**
>
> Thank you for taking the time to review our work and for your thoughtful feedback. We have submitted a revised version of our manuscript, in which all modifications to the main text are highlighted in blue.
>
> **Q1：Often the paper does not report error bars on their metrics, making it difficult to ascertain the significance of the results.**
>
> **A1:**
> We report uncertainty with error bars (mean ± standard deviation) for MSE, SSIM, LPIPS, and IS in Tables 2, 3, 6, 7, and 8. Because FID is defined over empirical image distributions rather than per-sample quantities, we provide FID as point estimates without error bars.
>
> **Q2：On the Mardia test, there is no adjustment for multiple testing.**
>
> **A2:**
> Actually, the Mardia test has adjustment for multiple testing. The package of R mardia{mvnormalTest} computes Mardia (1970)'s multivariate skewness and kurtosis statistics as well as their corresponding p-values.
>
> **Q3：Regarding the sampling from AE latent spaces -the authors use k-means + sampling. This can be susceptible to poor initializations - can the authors investigate more how this affects results for both AE and AE+GR?**
>
> **A3:**
> We sample the AE latent variables with different random seeds and apply k-means clustering under identical settings, then quantitatively evaluate the AE+GR generations. The results are reported in **Section 4.5.3 and Appendix F**. Across different initializations, AE+GR remains stable, as reflected by consistent IS and FID values.
>
> **Q4：Regarding diffusions - the diffusions are trained based on the fixed latent spaces learned by AE or AE+GR. Are the metrics computed over many different initializations and trainings of the AEs and AE+GR?**
>
> **A4:**
> Diffusion operates on a fixed latent space learned by an AE or AE+GR. Latent Diffusion Models are trained in two stages: (1) training the autoencoder and (2) training the diffusion model in the latent space. We apply our Gaussian constraint only in the first stage to strengthen latent distribution, and follow the original protocol for the second stage. We did not compute metrics across multiple random initializations or independent training runs.
>
> **Q5：I am confused how Gaussian regularization helps VQ-VAEs. VQ-VAEs have discrete latent spaces, not continuous. Can the authors explain?**
>
> **A5:**
> We believe there are two reasons. First, the VQ-VAEs codebook is learned from the latent variables, and a more regular latent space with a specified probabilistic distribution is easier for the codebook to learn. Second, both the original VQ-VAEs and its variants initialize the codebook with random Gaussian vectors, a simple and effective scheme. Aligning one Gaussian distribution with another Gaussian is easier than other distributions.
>
> **Q6：How is lambda selected?**
>
> **A6:**
> We set $\lambda=0.5$ for most autoencoder experiments and analyze its effect in **Section 4.5.1 and Appendix D**. Very small regularization strengths ($\lambda \le 0.05$) slightly improve reconstruction fidelity but noticeably reduce generation quality. Increasing $\lambda$ to around $0.1$ improves IS and lowers FID, with the best trade-off observed for $\lambda \in [0.1, 0.5]$. For excessively large values ($\lambda \ge 5$), both reconstruction and generation performance degrade due to over-regularization.
>
> **Q7：Can the authors add more experiments on how sensitive the method is to batch effects?**
>
> **A7:**
> We add some experiments on how sensitive the method is to batch effects in **Section 4.5.2 and Appendix E**. The results shows that batch size has little effect for AE+GReg.
>
> **Q8：Theoretical Explanation**
> **A8:**
> We propose the Gaussian constraint via random projections, motivated by the closure under linear transformations and the difficulty of testing normality in high dimensions. A standard way to analyze high-dimensional distributions is to project them to lower dimensions. So, we use linear random projections. Although our method may seem similar to the marginal skewness constraint, it is not the same. Calculating the marginal skewness needs first averaging the skewness of each component over the sample, and then averaging the resulting values. This is different from our method.

---

### Review · Reviewer_Yy66 · 2025-08-06

**Summary Of Contributions:**

The paper introduces a novel regularization method derived from the random projection theory to improve the underlying representation of a latent space of encoder-based models (like VAEs, AEs, GANs) to closely match multivariate Gaussian distribution. Unlike KL-divergence, which in case of VAEs assume independence amongst latent vectors, the proposed method does not make such an assumption. Moreover, the proposed method can evaluate distributional properties without explicit density modeling.

**Strengths:**

- The proposed regularization term (GReg) seems logical and explainable. I can see that this aligns more closely with the properties of how we would expect an underlying latent space to be.
- GReg is compatible with continuous as well as discrete latent spaces.

**Weakness:**

- My biggest concern comes from the qualitiative results. Despite following a logical storyline around matching multivariate Gaussian (without assuming independence or estimating the density) - the difference in the qualitative results are absolutely impercievable with and without GReg.  Even quantitatively, the gains seem marginal. This is not unexpected, as many of the current methods have been successful because they are simply "simple" despite having more complex methods in place showcasing marginal gains.
  - Fig 2: the authors claim an improvement in "scene layout", which is quite arbitrary - but I thought the results without GReg looked more realistic.
  - Fig 3: the identity and the attribute changes look the same with and without GReg.

- I understand the desire for such a term (GReg), but I am unable to map the gain in performance with any one very particular insight. It seems like, just in general, regularization improves performance - need not be GReg. For instance, in Fig 2, GReg improves the performance for AE (which does not come with another kind of regularization). But against VAE (that has KL as a regularizer), it has absolutely no improvement. As GReg is applied to LDM or VQVAE-2, I wonder if there are better methods to regularize these encoders to see similar gains.

- The authors perform the Mardia Test that implies improved multivariate Gaussian distribution - but I am again at loss on do we even need the space to be this accurate for the purposes of generative modeling if it does not improve the ultimate results?

**Audience:**

No

**Audience Explanation:**

The paper presents a logical argument using a theoritical concepts, but ultimately it struggles to showcase any substantial downstream improvement in results, which makes the whole premise questionable.

**Broader Impact Concerns:**

No concerns

**Claims And Evidence:**

No

**Claims Explanation:**

The experiments are quite insufficient:

- The first question I have is: what leads to a gain in performance? From the experiments, it seems like it is not sufficient to prove that this is by any means a superior form of regularization than all the other types of regularization. For example: in Table 2, GReg does not improve on VAE (with KL regularization).
- The qualitative results have created more doubts than convince me: some of the results without GReg seems to be better. The amount of qualitative resutls are not sufficient, and I would prefer a more rigorous human-evaulation to see if there's a general consensus on the qualtitative results.

**Requested Changes:**

I would like to see many more qualtitative results. A few pages of qualitiative results with systematic human evaluation (user study).

---

> ### Author Response · Authors · 2025-08-15
> **Response to Reviewer Yy66**
>
> Thank you for the thorough, detailed assessment of our work and your valuable suggestions. We agree that qualitative evaluation and clear evidence of where the gains come from are critical. We have submitted a revised version of our manuscript, in which all modifications to the main text are highlighted in blue.
>
> **Q1：Qualitative results are unconvincing; please add many more examples and a systematic human evaluation.**
>
> **A1：**
> **We have added more qualitative results.**
> We added two additional pages of examples for e4e editing (**Appendix G, Figs. 9–10**). We have also reported LPIPS, IS and FID for different editing (smile, age, pose) intensity in **Appendix I** as Additional quantitative evidence.
> **We have added a user study (100 respondents).**
> We conducted user studies for the qualitative evaluation of the generation effect of LDM and the inversion and editing (smile) effect of e4e. We found 100 respondents to complete both studies. The respondents are given a side-by-side comparison of images generated, reconstructed or edited by baseline and its Gaussian-regularized counterpart, and are asked to choose the more realistic image. Across both tasks, +GReg is preferred with statistical significance. Detailed data are reported in **Sec. 4.4 and Appendix H**.
> **We have done some clarity edits.** We removed subjective phrasing such as “improvement in scene layout.”
>
> **Q2：What leads to a gain in performance? From the experiments, it seems like it is not sufficient to prove that this is by any means a superior form of regularization than all the other types of regularization. For example: in Table 2, GReg does not improve on VAE (with KL regularization).**
>
>
> **A2：**
> We additionally compare our method against other Gaussian regularization. In high-dimensional settings, enforcing Gaussian is challenging. A common approach penalizes skewness and (excess) kurtosis to push each marginal toward a normal distribution. We implement these baselines on an autoencoder and benchmark them against our method. The results are reported in **Table 2**. Our approach consistently yields higher image quality, outperforming skewness and kurtosis penalties.
> **Why the gains on VAEs are limited:** VAEs already impose a strong latent constraint via the KL divergence to a $\mathcal{N}(0, I)$ prior. This leaves little room for additional regularization to reshape the latent distribution, and extra constraints can even conflict with the ELBO trade-off. As a result, our method brings little improvement in the VAE setting.
>
> **Q3：The authors perform the Mardia Test that implies improved multivariate Gaussian distribution - but I am again at loss on do we even need the space to be this accurate for the purposes of generative modeling if it does not improve the ultimate results?**
>
> **A3**
> **Why the gains on LDMs are modest.**
> On latent-diffusion models (LDMs), the improvement is smaller because our regularization is mild and does not directly optimize the final samples. The state-of-the-art generators already produce very strong outputs, so it is hard to improve them by a mild way. The value of our method lies in its broad applicability. It is lightweight, model-agnostic, and consistently yields improvements across a wide range of encoder-based architectures.
> **Advantages of enforcing a Gaussian latent space.** Imposing a probabilistic Gaussian prior on the latent space supports precise sampling and related operations. It also benefits image editing: many editing techniques are linear (or locally affine), and Gaussians are closed under affine transformations. The detailed discussion is in Section 4.3.3.

---

> > ### Comment · Reviewer_Yy66 · 2025-09-22
> > **Response to the rebuttal**
> >
> > I thank the authors for the response and I have gone through the response along with the additional qualitiative results, however, I am still unsatisfied with the results as I find the proposed results very obviously indistinguishable from the results of the previous works. I appreciate the user study, however, I still stand by my initial assesment.

---

### Author Response · Authors · 2025-09-11

Dear Reviewers,

Thank you again for your valuable time and feedback on our paper. We have submitted our revision and just wanted to kindly check if you have any further questions or comments.

Best regards,

Authors

---

### Decision · Action_Editor_QVM7 · 2025-10-17

**Recommendation:** Reject

**Audience:**

Yes

**Audience Explanation:**

Better regularisation in latent spaces is of interest to the community working on autoencoders, VAEs, diffusion models, and generative modeling more broadly.

**Claims And Evidence:**

No

**Claims Explanation:**

While the paper is well written and the proposed method is interesting, the experimental evidence remains weak and does not convincingly demonstrate clear improvements over established baselines. Two of the three reviewers strongly questioned the significance of the empirical contribution, and their concerns persisted even after the rebuttal. Though one reviewer was mildly positive, the overall balance of reviews clearly leans negative. Given the expectations of TMLR, I recommend rejection at this stage.